# Explicit Disentanglement of Appearance and Perspective in Generative Models

**Nicki S. Detlefsen** *
nsde@dtu.dk

**Søren Hauberg** *
sohau@dtu.dk

## Abstract

Disentangled representation learning finds compact, independent and easy-to-interpret factors of the data. Learning such has been shown to require an inductive bias, which we explicitly encode in a generative model of images. Specifically, we propose a model with two latent spaces: one that represents spatial transformations of the input data, and another that represents the transformed data. We find that the latter naturally captures the intrinsic appearance of the data. To realize the generative model, we propose a Variationally Inferred Transformational Autoencoder (VITAE) that incorporates a spatial transformer into a variational autoencoder. We show how to perform inference in the model efficiently by carefully designing the encoders and restricting the transformation class to be diffeomorphic. Empirically, our model separates the visual style from digit type on MNIST, separates shape and pose in images of human bodies and facial features from facial shape on CelebA.

## 1 Introduction

*Disentangled Representation Learning (DRL)* is a fundamental challenge in machine learning that is currently seeing a renaissance within deep generative models. DRL approaches assume that an AI agent can benefit from separating out (disentangle) the underlying structure of data into disjointed parts of its representation. This can furthermore help interpretability of the decisions of the AI agent and thereby make them more accountable.

Even though there have been attempts to find a single formalized notion of disentanglement [Higgins et al., 2018], no such theory exists (yet) which is widely accepted. However, the intuition is that a disentangled representation $z$ should separate different informative factors of variation in the data [Bengio et al., 2012]. This means that changing a single latent dimension $z_i$ should only change a single interpretable feature in the data space $\mathcal{X}$.

Within the DRL literature, there are two main approaches. The first is to hard-wire disentanglement into the model, thereby creating an inductive bias. This is well known *e.g.* in convolutional neural networks, where the convolution operator creates an inductive bias towards translation in data. The second approach is to instead learn a representation that is faithful to the underlying data structure, hoping that this is sufficient to disentangle the representation. However, there is currently little to no agreement in the literature on how to learn such representations [Locatello et al., 2019].

We consider disentanglement of two explicit groups of factors, the *appearance* and the *perspective*. We here define the appearance as being the factors of data that are left after transforming $x$ by its perspective. Thus, the appearance is the *form* or *archetype* of an object and the perspective represents the specific realization of that archetype. Practically speaking, the perspective could correspond to an image rotation that is deemed irrelevant, while the appearance is a representation of the rotated image, which is then invariant to the perspective. This interpretation of the world goes back to Plato's allegory of the cave, from which we also borrow our terminology. This notion of removing

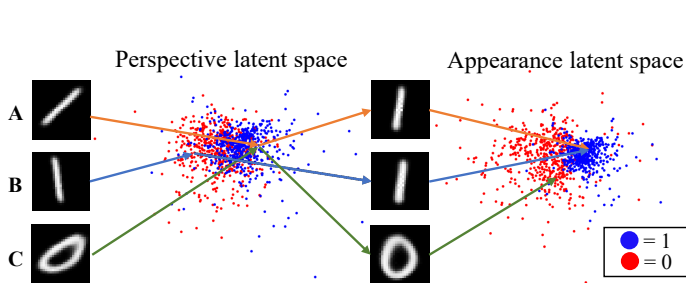

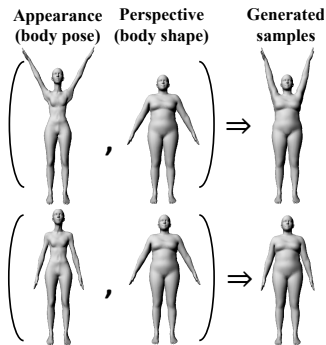

**Figure 1:** We disentangle data into *appearance* and *perspective* factors. First, data are encoded based on their *perspective* (in this case image A and C are rotated in the same way), which is then removed from the original input. Hereafter, the transformed samples can be encoded in the *appearance* space (image A and B are both ones), that encodes the factors left in data.

**Figure 2:** Our model, VITAE, disentangles appearance from perspective. Here we separate body pose (arm position) from body shape.

perspective before looking at the appearance is well-studied within supervised learning, *e.g.* using *spatial transformer nets (STNs)* [Jaderberg et al., 2015].

**This paper contributes** an explicit model for disentanglement of appearance and perspective in images, called the *variational inferred transformational autoencoder (VITAE)*. As the name suggests, we focus on variational autoencoders as generative models, but the idea is general (Fig. 1). First we encode/decode the perspective features in order to extract an appearance that is perspective-invariant. This is then encoded into a second latent space, where inputs with similar appearance are encoded similarly. This process generates an inductive bias that disentangles perspective and appearance. In practice, we develop an architecture that leverages the inference part of the model to guide the generator towards better disentanglement. We also show that this specific choice of architecture improves training stability with the right choice of parametrization of perspective factors. Experimentally, we demonstrate that our model on four datasets: standard disentanglement benchmark dSprites, disentanglement of style and content on MNIST, pose and shape on images of human bodies (Fig. 2) and facial features and facial shape on CelebA.

## 2 Related work

**Disentangled representations learning (DRL)** have long been a goal in data analysis. Early work on *non-negative matrix factorization* [Lee and Seung, 1999] and *bilinear models* [Tenenbaum and Freeman, 2000] showed how images can be composed into semantic "parts" that can be glued together to form the final image. Similarly, *EigenFaces* [Turk and Pentland, 1991] have often been used to factor out lighting conditions from the representation [Shakunaga and Shigenari, 2001], thereby discovering some of the physics that govern the world of which the data is a glimpse. This is central in the long-standing argument that for an AI agent to understand and reason about the world, it must disentangle the explanatory factors of variation in data [Lake et al., 2016]. As such, DRL can be seen as a poor man's approximation to discovering the underlying causal factors of the data.

**Independent components** are, perhaps, the most stringent formalization of "disentanglement". The seminal *independent component analysis (ICA)* [Comon, 1994] factors the signal into statistically independent components. It has been shown that the independent components of *natural images* are edge filters [Bell and Sejnowski, 1997] that can be linked to the receptive fields in the human brain [Olshausen and Field, 1996]. Similar findings have been made for both *video* and *audio* [van Hateren and Ruderman, 1998, Lewicki, 2002]. DRL, thus, allows us to understand both the data and ourselves. Since independent factors are the optimal compression, ICA finds the most compact representation, implying that the predictive model can achieve maximal capacity from its parameters. This gives DLR a predictive perspective, and can be taken as a hint that a well-trained model might be disentangled. In

the linear case, independent components have many successful realizations [Hyvärinen and Oja, 2000], but in the general non-linear case, the problem is not identifiable [Hyvärinen et al., 2018].

**Deep DRL** was initiated by Bengio et al. [2012] who sparked the current interest in the topic. One of the current state-of-the-art methods for doing disentangled representation learning is the $\beta$-VAE [Higgins et al., 2017], that modifies the *variational autoencoder (VAE)* [Kingma and Welling, 2013, Rezende et al., 2014] to learn a more disentangled representation. $\beta$-VAE enforces more weight on the KL-divergence in the VAE loss, thereby optimizing towards latent factors that should be axis aligned *i.e.* disentangled. Newer models like $\beta$-TCVAE [Chen et al., 2018] and DIP-VAE [Kumar et al., 2017] extend $\beta$-VAE by decomposing the KL-divergences into multiple terms, and only increase the weight on terms that analytically disentangles the models. InfoGAN [Chen et al., 2016] extends the latent code $\boldsymbol{z}$ of the standard GAN model [Goodfellow et al., 2014] with an extra latent code $c$ and then penalize low mutual information between generated samples $G(c, z)$ and $c$. DC-IGN [Kulkarni et al., 2015] forces the latent codes to be disentangled by only feeding in batches of data that vary in one way (*e.g.* pose, light) while only having small disjoint parts of the latent code active.

**Shape statistics** is the key inspiration for our work. The shape of an object was first formalized by Kendall [1989] as being what is left of an object when *translation*, *rotation* and *scale* are factored out. That is, the intrinsic shape of an object should not depend on viewpoint. This idea dates, at least, back to D'Arcy Thompson [1917] who pioneered the understanding of the development of biological forms. In Kendall's formalism, the rigid transformations (translation, rotation and scale) are viewed as group actions to be factored out of the representation, such that the remainder is *shape*. Higgins et al. [2018] follow the same idea by defining disentanglement as a factoring of the representation into group actions. Our work can be seen as a realization of this principle within a deep generative model. When an object is represented by a set of landmarks, *e.g.* in the form of discrete points along its contour, then Kendall's *shape space* is a Riemannian manifold that exactly captures all variability among the landmarks except translation, rotation, and scale of the object. When the object is not represented by landmarks, then similar mathematical results are not available. Our work shows how the same idea can be realized for general image data, and for a much wider range of transformations than the rigid ones. Learned-Miller [2006] proposed a related linear model that generate new data by transforming a prototype, which is estimated by joint alignment.

**Transformations** are at the core of our method, and these leverage the architecture of spatial transformer nets (STNs) [Jaderberg et al., 2015]. While these work well within supervised learning, [Lin and Lucey, 2016, Annunziata et al., 2018, Detlefsen et al., 2018] there has been limited uptake within generative models. Lin et al. [2018] combine a GAN with an STN to compose a foreground (e.g a furniture) into a background such that it look neutral. The AIR model [Eslami et al., 2016] combines STNs with a VAE for object rendering, but do not seek disentangled representations. In supervised learning, *data augmentation* is often used to make a classifier partially invariant to select transformations [Baird, 1992, Hauberg et al., 2016].

## 3 Method

Our goal is to extend a variational autoencoder (VAE) [Kingma and Welling, 2013, Rezende et al., 2014] such that it can disentangle appearance and perspective in data. A standard VAE assumes that data is generated by a set of latent variables following a standard Gaussian prior,

$$p(\boldsymbol{x}) = \int p(\boldsymbol{x}|\boldsymbol{z})p(\boldsymbol{z})\mathrm{d}\boldsymbol{z}$$

$$p(\boldsymbol{z}) = \mathcal{N}(\boldsymbol{0}, \mathbb{I}_d), \ p(\boldsymbol{x}|\boldsymbol{z}) = \mathcal{N}(\boldsymbol{x}|\boldsymbol{\mu}_p(\boldsymbol{z}), \boldsymbol{\sigma}_p^2(\boldsymbol{z})) \ \text{or} \ P(\boldsymbol{x}|\boldsymbol{z}) = \mathcal{B}(\boldsymbol{x}|\boldsymbol{\mu}_p(\boldsymbol{z})). \tag{1}$$

Data $\boldsymbol{x}$ is then generated by first sampling a latent variable $\boldsymbol{z}$ and then sample $\boldsymbol{x}$ from the conditional $p(\boldsymbol{x}|\boldsymbol{z})$ (often called the decoder). To make the model flexible enough to capture complex data distributions, $\boldsymbol{\mu}_p$ and $\boldsymbol{\sigma}_p^2$ are modeled as deep neural nets. The marginal likelihood is then intractable and a variational approximation $q$ to $p(\boldsymbol{z}|\boldsymbol{x})$ is needed,

$$p(\boldsymbol{z}|\boldsymbol{x}) \approx q(\boldsymbol{z}|\boldsymbol{x}) = \mathcal{N}(\boldsymbol{z}|\boldsymbol{\mu}_q(\boldsymbol{x}), \boldsymbol{\sigma}_q^2(\boldsymbol{x})), \tag{2}$$

where $\boldsymbol{\mu}_q(\boldsymbol{x})$ and $\boldsymbol{\sigma}_q^2(\boldsymbol{x})$ are deep neural networks, see Fig. 3(a).

When training VAEs, we therefore simultaneously train a generative model $p_\theta(\boldsymbol{x}|\boldsymbol{z})p_\theta(\boldsymbol{z})$ and an inference model $q_\phi(\boldsymbol{z}|\boldsymbol{x})$ (often called the encoder). This is done by maximizing a variational lower

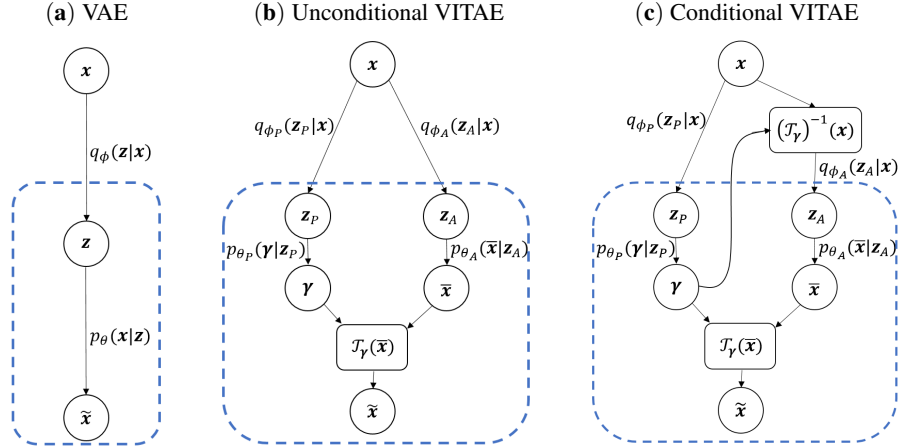

**Figure 3:** Architectures of standard VAE and our proposed U-VITAE and C-VITAE models. Here $q$ denotes encoders, $p$ denotes decoders, $\mathcal{T}^\gamma$ denotes a ST-layer with transformation parameters $\gamma$. The dotted box indicates the generative model.

bound to the likelihood $p(\boldsymbol{x})$ called the *evidence lower bound (ELBO)*

$$\log p(\boldsymbol{x}) \geq \mathbb{E}_{q_\phi(\boldsymbol{z}|\boldsymbol{x})}\left[\log \frac{p_\theta(\boldsymbol{x},\boldsymbol{z})}{q_\phi(\boldsymbol{z}|\boldsymbol{x})}\right] = \underbrace{\mathbb{E}_{q_\phi(\boldsymbol{z}|\boldsymbol{x})}\left[\log p_\theta(\boldsymbol{x}|\boldsymbol{z})\right]}_{\text{data fitting term}} - \underbrace{KL(q_\phi(\boldsymbol{z}|\boldsymbol{x})||p_\theta(\boldsymbol{z}))}_{\text{regulazation term}}. \quad (3)$$

The first term measures the reconstruction error between $\boldsymbol{x}$ and $p_\theta(\boldsymbol{x}|\boldsymbol{z})$ and the second measures the KL-divergence between the encoder $q_\phi(\boldsymbol{z}|\boldsymbol{x})$ and the prior $p(\boldsymbol{z})$. Eq. 3 can be optimized using the reparametrization trick [Kingma and Welling, 2013]. Several improvements to VAEs have been proposed [Burda et al., 2015, Kingma et al., 2016], but our focus is on the standard model.

## 3.1 Incorporating an inductive bias

To incorporate an inductive bias that is able to disentangle appearance from perspective, we change the underlying generative model to rely on two latent factors $\boldsymbol{z}_A$ and $\boldsymbol{z}_P$,

$$p(\boldsymbol{x}) = \iint p(\boldsymbol{x}|\boldsymbol{z}_A, \boldsymbol{z}_P)p(\boldsymbol{z}_A)p(\boldsymbol{z}_P)\mathrm{d}\boldsymbol{z}_A\mathrm{d}\boldsymbol{z}_P, \quad (4)$$

where we assume that $\boldsymbol{z}_A$ and $\boldsymbol{z}_P$ both follow standard Gaussian priors. Similar to a VAE, we also model the generators as deep neural networks. To generate new data $\boldsymbol{x}$, we combine the appearance and perspective factors using the following 3-step procedure that uses a spatial transformer (ST) layer [Jaderberg et al., 2015] (dotted box in Fig. 3(b)):

1. Sample $\boldsymbol{z}_A$ and $\boldsymbol{z}_P$ from $p(\boldsymbol{z}) = \mathcal{N}(\boldsymbol{0}, \mathbb{I}_d)$.

2. Decode both samples $\tilde{\boldsymbol{x}} \sim p(\boldsymbol{x}|\boldsymbol{z}_A)$, $\boldsymbol{\gamma} \sim p(\boldsymbol{x}|\boldsymbol{z}_P)$.

3. Transform $\tilde{\boldsymbol{x}}$ with parameters $\boldsymbol{\gamma}$ using a spatial transformer layer: $\boldsymbol{x} = \mathcal{T}_\gamma(\tilde{\boldsymbol{x}})$.

This process is illustrated by the dotted box in Fig. 3(b).

**Unconditional VITAE inference.** As the marginal likelihood (4) is intractable, we use variational inference. A natural choice is to approximate each latent group of factors $\boldsymbol{z}_A, \boldsymbol{z}_P$ independently of the other *i.e.*

$$p(\boldsymbol{z}_P|\boldsymbol{x}) \approx q_P(\boldsymbol{z}_P|\boldsymbol{x}) \text{ and } p(\boldsymbol{z}_A|\boldsymbol{x}) \approx q_A(\boldsymbol{z}_A|\boldsymbol{x}). \quad (5)$$

The combined inference and generative model is illustrated in Fig. 3(b). For comparison, a VAE model is shown in Fig. 3(a). It can easily be shown that the ELBO for this model is merely a VAE with a KL-term for each latent space (see supplements).

**Conditional VITAE inference.** This inference model does *not* mimic the generative process of the model, which may be suboptimal. Intuitively, we expect the encoder to approximately perform the inverse operation of the decoder, *i.e.* $\boldsymbol{z} \approx \text{encoder}(\text{decoder}(\boldsymbol{z})) \approx \text{decoder}^{-1}(\text{decoder}(\boldsymbol{z}))$. Since the proposed encoder (5) does not include an ST-layer, it may be difficult to train an encoder to approximately invert the decoder. To accommodate this, we first include an ST-layer in the encoder for the appearance factors. Secondly, we explicitly enforce that the predicted transformation in the encoder $\mathcal{T}^{\gamma_e}$ is the inverse of that of the decoder $\mathcal{T}^{\gamma_d}$, *i.e.* $\mathcal{T}^{\gamma_e} = (\mathcal{T}^{\gamma_d})^{-1}$ (more on invertibility in Sec. 3.2). The inference of appearance is now dependent on the perspective factor $\boldsymbol{z}_P$, *i.e.*

$$p(\boldsymbol{z}_P|\boldsymbol{x}) \approx q_P(\boldsymbol{z}_P|\boldsymbol{x}) \text{ and } p(\boldsymbol{z}_A|\boldsymbol{x}) \approx q_A(\boldsymbol{z}_A|\boldsymbol{x}, \boldsymbol{z}_P). \tag{6}$$

These changes to the inference architecture are illustrated in Fig. 3(c). It can easily be shown that the ELBO for this model is given by

$$\log p(\boldsymbol{x}) \geq \mathbb{E}_{q_A, q_P}[\log(p(\boldsymbol{x}|\boldsymbol{z}_A, \boldsymbol{z}_P)] - D_{KL}(q_P(\boldsymbol{z}_P|\boldsymbol{x})||p(\boldsymbol{z}_P)) - \mathbb{E}_{q_P}[D_{KL}(q_A(\boldsymbol{z}_A|\boldsymbol{x})||p(\boldsymbol{z}_A))]. \tag{7}$$

which resembles the standard ELBO with a additional term (derivation in supplementary material), corresponding to the second latent space. We will call both models *variational inferred transformational autoencoders (VITAE)* and we will denote the first model (5) as *unconditional/U-VITAE* and the second model (6) as *conditional/C-VITAE*. The naming comes from Eq. 5 and 6, where $\boldsymbol{z}_A$ is respectively unconditioned and conditioned on $\boldsymbol{z}_P$. Experiments will show that the conditional architecture is essential for inference (Sec. 4.2).

## 3.2 Transformation classes

Until now, we have assumed that there exists a class of transformations $\mathcal{T}$ that captures the perspective factors in data. Clearly, the choice of $\mathcal{T}$ depends on the true factors underlying the data, but in many cases an affine transformation should suffice.

$$\mathcal{T}_{\boldsymbol{\gamma}}(\boldsymbol{x}) = \mathbf{A}\boldsymbol{x} + \mathbf{b} = \begin{bmatrix} \gamma_{11} & \gamma_{12} & \gamma_{13} \\ \gamma_{21} & \gamma_{22} & \gamma_{14} \end{bmatrix} \begin{bmatrix} x \\ y \\ 1 \end{bmatrix}. \tag{8}$$

However, the C-VITAE model requires access to the inverse transformation $\mathcal{T}^{-1}$. The inverse of Eq. 8 is given by $\mathcal{T}_{\boldsymbol{\gamma}}^{-1}(\boldsymbol{x}) = \mathbf{A}^{-1}\boldsymbol{x} - \mathbf{b}$, which only exist if $\mathbf{A}$ has a non-zero determinant.

One, easily verified, approach to secure invertibility is to parametrize the transformation by two scale factors $s_x, s_y$, one rotation angle $\alpha$, one shear parameter $m$ and two translation parameters $t_x, t_y$:

$$\mathcal{T}_{\boldsymbol{\gamma}}(\boldsymbol{x}) = \begin{bmatrix} \cos(\alpha) & -\sin(\alpha) \\ \sin(\alpha) & \cos(\alpha) \end{bmatrix} \begin{bmatrix} 1 & m \\ 0 & 1 \end{bmatrix} \begin{bmatrix} s_x & 0 \\ 0 & s_y \end{bmatrix} + \begin{bmatrix} t_x \\ t_y \end{bmatrix}. \tag{9}$$

In this case the inverse is trivially

$$\mathcal{T}_{(s_x, s_y, \gamma, m, t_x, t_y)}^{-1}(\boldsymbol{x}) = \mathcal{T}_{(\frac{1}{s_x}, \frac{1}{s_y}, -\gamma, -m, -t_x, -t_y)}(\boldsymbol{x}), \tag{10}$$

where the scale factors must be strictly positive.

An easier and more elegant approach is to leverage the matrix exponential. That is, instead of parametrizing the transformation in Eq. 8, we instead parametrize the velocity of the transformation

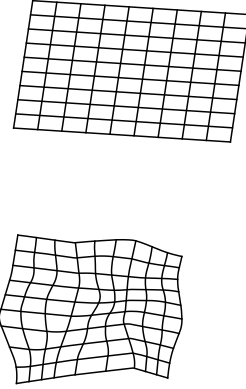

**Figure 4:** Random deformation field of an affine transformation (top) compared to a CPAB (bottom). We clearly see that CPAB transformations offers a mush more flexible and rich class of diffiomorphic transformations.

$$\mathcal{T}_{\boldsymbol{\gamma}}(\boldsymbol{x}) = \mathbf{expm}\left( \begin{bmatrix} \gamma_{11} & \gamma_{12} & \gamma_{13} \\ \gamma_{21} & \gamma_{22} & \gamma_{14} \\ 0 & 0 & 0 \end{bmatrix} \right) \begin{bmatrix} x \\ y \\ 1 \end{bmatrix}. \tag{11}$$

The inverse[2] is then $\mathcal{T}_{\boldsymbol{\gamma}}^{-1} = \mathcal{T}_{-\boldsymbol{\gamma}}$. Then $\mathcal{T}$ in Eq. 11 is a $C^\infty$-diffiomorphism (*i.e.* a differentiable invertible map with a differentiable inverse) [Duistermaat and Kolk, 2000]. Experiments show that diffeomorphic transformations stabilize training and yield tighter ELBOs (see supplements).

Often we will not have prior knowledge regarding which transformation classes are suitable for disentangling the data. A natural way forward is then to apply a highly flexible class of transformations that are treated as "black-box". Inspired by Detlefsen et al. [2018], we also consider transformations $\mathcal{T}_\gamma$ using the highly expressive diffiomorphic transformations *CPAB* from Freifeld et al. [2015]. These can be viewed as an extension to Eq. 11: instead of having a single affine transformation parametrized by its velocity, the image domain is divided into smaller cells, each having their own affine velocity. The collection of local affine velocities can be efficiently parametrized and integrated, giving a fast and flexible diffeomorphic transformation, see Fig. 4 for a comparison between an affine transformation and a CPAB transformation. For details, see Freifeld et al. [2015].

We note, that our transformer architecture are similar to the work of Lorenz et al. [2019] and Xing et al. [2019] in that they also tries to achieve disentanglement through spatial transformations. However, our work differ in the choice of transformation. This is key, as the theory of Higgins et al. [2018] strongly relies on disentanglement through *group actions*. This places hard constrains on which spatial transformations are allowed: *they have to form a smooth group*. Both thin-plate-spline transformations considered in Lorenz et al. [2019] and displacement fields considered in Xing et al. [2019] are not invertible and hence do not correspond to proper group actions. Since diffiomorphic transformations form a smooth group, this choice is paramount to realize the theory of Higgins et al. [2018].

## 4 Experimental results and discussion

For all experiments, we train a standard VAE, a $\beta$-VAE [Higgins et al., 2017], a $\beta$-TCVAE [Chen et al., 2018], a DIP-VAE-II [Kumar et al., 2017] and our developed VITAE model. We model the encoders and decoders as multilayer perceptron networks (MLPs). For a fair comparison, the number of trainable parameters is approximately the same in all models. The models were implemented in Pytorch [Paszke et al., 2017] and the code is available at `https://github.com/SkafteNicki/unsuper/`.

**Evaluation metric.** Measuring disentanglement still seems to be an unsolved problem, but the work of Locatello et al. [2019] found that most proposed disentanglement metrics are highly correlated. We have chosen to focus on the DIC-metric from Eastwood and Williams [2019], since this metric has seen some uptake in the research community. This metric measures how will the generative factors can be predicted from latent factors. For the MNIST and SMPL datasets, the generative factors are discrete instead of continuous, so we change the standard linear regression network to a kNN-classification algorithm. We denote this metric $D_{score}$ in the results.

### 4.1 Disentanglement on shapes

We initially test our models on the dSprites dataset [Matthey et al., 2017], which is a well established disentanglement benchmarking dataset to evaluate the performance of disentanglement algorithms. The results can be seen in Table 1. We find that our proposed C-VITAE model perform best, followed

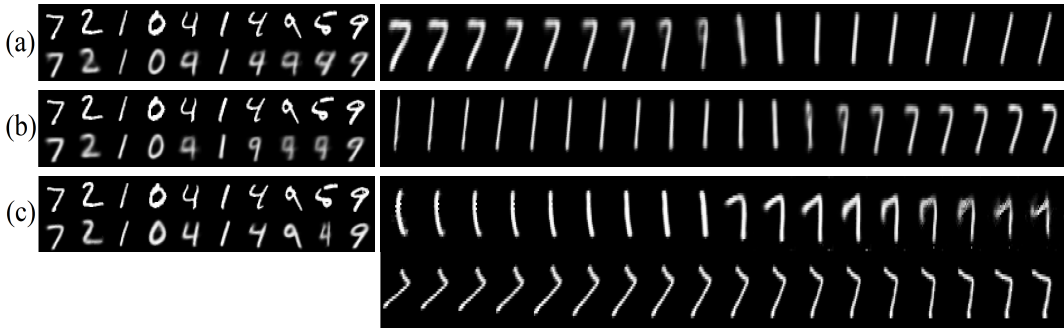

**Figure 5:** Reconstructions (left images) and manipulation of latent codes (right images) on MNIST for the three different models: VAE (a), $\beta$-VAE (b) and C-VITAE (c). The right images are generated by varying one latent dimension in all models, while keeping the rest fixed. For the C-VITAE model, we have shown this for both the appearance and perspective spaces.

| | dSprite | | | MNIST | | | SMPL | | |
|---|---|---|---|---|---|---|---|---|---|
| | ELBO | $\log p(\boldsymbol{x})$ | $D_{score}$ | ELBO | $\log p(\boldsymbol{x})$ | $D_{score}$ | ELBO | $\log p(\boldsymbol{x})$ | $D_{score}$ |
| VAE | -47.05 | -49.32 | 0.05 | -169 | -172 | 0.579 | $-8.62 \times 10^3$ | $-8.62 \times 10^3$ | 0.485 |
| $\beta$-VAE | -79.45 | -81.38 | 0.18 | -150 | -152 | 0.653 | $-8.62 \times 10^3$ | $-8.60 \times 10^3$ | 0.525 |
| $\beta$-TCVAE | -66.48 | -68.12 | 0.30 | -141 | -144 | 0.679 | $-8.62 \times 10^3$ | $-8.56 \times 10^3$ | 0.651 |
| DIP-VAE-II | **-46.32** | **-48.92** | 0.12 | -140 | -155 | 0.733 | $-8.62 \times 10^3$ | $-8.54 \times 10^3$ | 0.743 |
| U-VITAE | -55.25 | -57.29 | 0.22 | -142 | -143 | 0.782 | $-8.62 \times 10^3$ | $-8.55 \times 10^3$ | 0.673 |
| C-VITAE | -68.26 | -70.49 | **0.38** | **-139** | **-141** | **0.884** | $-8.62 \times 10^3$ | $-8.52 \times 10^3$ | **0.943** |

**Table 1:** Quantitative results on three datasets. For each dataset we report the ELBO, test set log likelihood and disentanglement score $D_{score}$. Bold marks best results.

by the $\beta$-TCVAE model in terms of disentanglement. The experiments clearly shows the effect on performance of the improved inference structure of C-VITAE compared to U-VITAE. It can be shown that the conditional architecture of C-VITAE, minimizes the mutual information between $\boldsymbol{z}_A$ and $\boldsymbol{z}_P$, leading to better disentanglement of the two latent spaces. To get the U-VITAE architecture to work similarly would require a auxiliary loss term added to the ELBO.

### 4.2 Disentanglement of MNIST images

Secondly, we test our model on the MNIST dataset [LeCun et al., 1998]. To make the task more difficult, we artificially augment the dataset by first randomly rotating each image by an angle uniformly chosen in the interval $[-20°, 20°]$ and secondly translating the images by $t = [x, y]$, where $x, y$ is uniformly chosen from the interval [-3, 3]. For VITAE, we model the perspective with an affine diffiomorphic transformation (Eq. 11).

The quantitative results can be seen in Table 1. We clearly see that C-VITAE outperforms the alternatives on all measures. We overall observes that better disentanglement, seems to give better distribution fitting. Qualitatively, Fig. 5 shows the effect of manipulating the latent codes alongside test reconstructions for VAE, $\beta$-VAE and C-VITAE. Due to space constraints, the results from $\beta$-TCVAE and DIP-VAE-II can found in the supplementary material. The plots were generated by following the protocol from Higgins et al. [2017]: one latent factor is linearly increased from -3 to 3, while the rest is kept fixed. In the VAE (Fig. 5(a)), this changes both the appearance (going from a 7 to a 1) and the perspective (going from rotated slightly left to rotated right). We see no meaningful disentanglement of latent factors. In the $\beta$-VAE model (Fig. 5(b)), we observe some disentanglement, since only the appearance changes with the latent factor. However this disentanglement comes at the cost of poor reconstructions. This trade-off is directly linked to the emphasized regularization in the $\beta$-VAE. We note that the value $\beta = 4.0$ proposed in the original paper [Higgins et al., 2017] is insufficiently low for our experiments to observe any disentanglement, and we use $\beta = 8.0$ based on qualitative evaluation of results. For $\beta$-TCVAE and DIP-VAE-II we observe nearly the same amount of qualitative disentanglement as $\beta$-VAE, however these models achieve less blurred samples and reconstructions. This is probably due to the two models decomposition of the KL-term, only increasing the parts that actually contributes to disentanglement. Finally, for our developed VITAE model (Fig. 5(c)), we clearly see that when we change the latent code in the appearance space (top row), we only change the content of the generated images, while manipulating the latent code in the perspective space (bottom row) only changes the perspective *i.e.* image orientation.

Interestingly, we observe that there exists more than one prototype of a 1 in the appearance space of VITAE, going from slightly bent to straightened out. By our definition of disentanglement, that *everything left* after transforming the image is appearance, there is nothing wrong with this. This is simply a consequence of using an affine transformation that cannot model this kind of local deformation. Choosing a more flexible transformation class could factor out this kind of perspective. The supplements contain generated samples from the different models.

### 4.3 Disentanglement of body shape and pose

We now consider synthetic image data of human bodies generated by the *Skinned Multi-Person Linear Model (SMPL)* [Loper et al., 2015] which are explicitly factored into *shape* and *pose*. We generate 10,000 bodies (8,000 for training, 2,000 for testing), by first continuously sampling body shape (going from thin to thick) and then uniformly sampling a body pose from four categories ((arms up, tight),

(arms up, wide), (arms down, tight), (arms down, wide)). Fig. 2 shows examples of generated images. Since change in body shape approximately amounts to a local shape deformation, we model the perspective factors using the aforementioned "black-box" diffiomorphic CPAB transformations (Sec. 3.2). The remaining appearance factor should then reflect body pose.

**Quantitative evaluation.** We again refer to Table 1 that shows ELBO, test set log-likelihood and disentanglement score for all models. As before, C-VITAE is both better at modelling the data distribution and achieves a higher disentanglement score. The explanation is that for a standard VAE model (or $\beta$-VAE and its variants for that sake) to learn a complex body shape deformation model, it requires a high capacity network. However, the VITAE architecture gives the autoencoder a short-cut to learning these transformations that only requires optimizing a few parameters. We are not guaranteed that the model will learn anything meaningful or that it actually uses this short-cut, but experimental evidence points in that direction. A similar argument holds in the case of MNIST, where a standard MLP may struggle to learn rotation of digits, but the ST-layer in the VITAE architecture provides a short-cut. Furthermore, we found the training of VITAE to be more stable than other models.

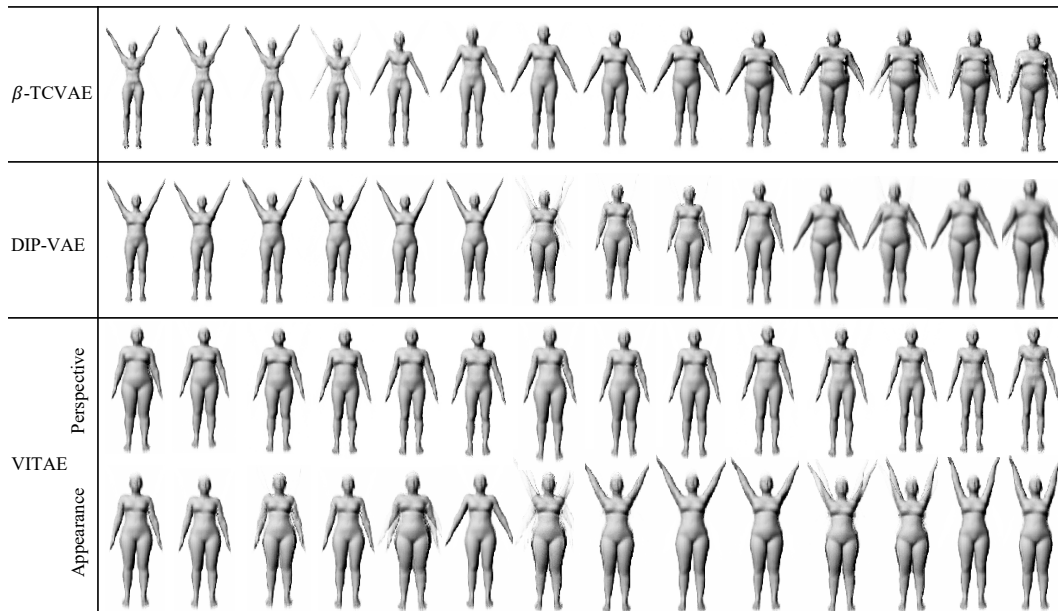

**Figure 6:** Disentanglement of body shape and body pose on SMPL-generated bodies for three different models. The images are generated by varying one latent dimension, while keeping the rest fixed. For the C-VITAE model we have shown this for both the appearance and perspective spaces, since this is the only model where we quantitatively observe disentanglement.

**Qualitative evaluation.** Again, we manipulate the latent codes to visualize their effect (Fig. 6). This time, we here show the result for $\beta$-TCVAE, DIP-VAE-II and VITAE. The results from standard VAE and $\beta$-VAE can be found in supplementary material. For both $\beta$-TCVAE and DIP-VAE-II we do not observe disentanglement of body pose and shape, since the decoded images both change arm position (from up to down) and body shape. We note that for both $\beta$-VAE, $\beta$-TCVAE and DIP-VAE-II we did a grid search for their respective hyper parameters. For these three models, we observe that the choice of hyper parameters (scaling of KL term) can have detrimental impact of reconstructions and generated samples. Due to lack of space, test set reconstructions and generated samples can be found in the supplementary material. For VITAE we observe some disentanglement of body pose and shape, as variation in appearance space mostly changes the positions of the arms, while the variations in the perspective space mostly changes body shape. The fact that we cannot achieve full disentanglement of this SMPL dataset indicates the difficulty of the task.

## 4.4 Disentanglement on CelebA

Finally, we qualitatively evaluated our proposed model on the CelebA dataset [Liu et al., 2015]. Since this is a " real life " dataset we do not have access to generative factors and we can therefore only qualitatively evaluate the model. We again model the perspective factors using the aforementioned CPAB transformations, which we assume can model the facial shape. The results can be seen in Fig. 7, which shows latent traversals of both the perspective and appearance factors, and how they influence the generated images. We do observe some interpolation artifacts that are common for architectures using spatial transformers.

(**a**) Changing $z_{P,1}$ corresponds to facial size.

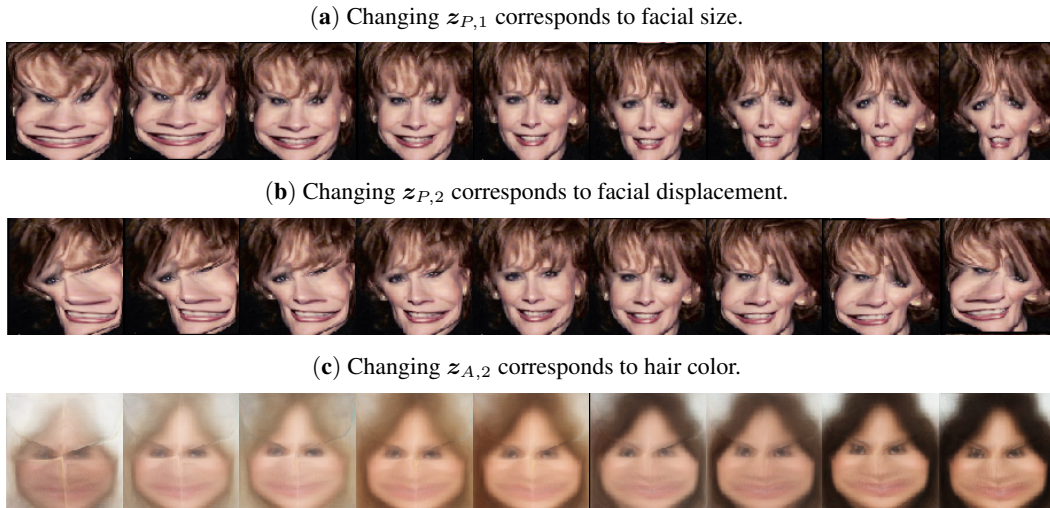

(**b**) Changing $z_{P,2}$ corresponds to facial displacement.

(**c**) Changing $z_{A,2}$ corresponds to hair color.

**Figure 7:** Traversal in latent space shows, that our model can disentangle complex factors such as facial size, facial position and hair color.

## 5 Summary

In this paper, we have shown how to explicitly disentangle *appearance* from *perspective* in a variational autoencoder [Kingma and Welling, 2013, Rezende et al., 2014]. This is achieved by incorporating a spatial transformer layer [Jaderberg et al., 2015] into both encoder and decoder in a coupled manner. The concepts of appearance and perspective are broad as is evident from our experimental results in human body images, where they correspond to *pose* and *shape*, respectively. By choosing the class of transformations in accordance with prior knowledge it becomes an effective tool for controlling the inductive bias needed for disentangled representation learning. On both MNIST and body images our method quantitatively and qualitatively outperforms general purpose disentanglement models [Higgins et al., 2017, Chen et al., 2018, Kumar et al., 2017]. We find it unsurprisingly that in situations where some prior knowledge about the generative factors is known, encoding these in the into the model give better result than ignoring such information.

Our results support the hypothesis [Higgins et al., 2018] that inductive biases are necessary for learning disentangled representations, and our model is a step in the direction of getting fully disentangled generative models. We envision that the VITAE model should be combined with other models, by first using the VITAE model to separate appearance and perspective, and then training a second model only on the appearance. This will factor out one latent factor at a time, leaving a hierachy of disentangled factors.

**Acknowledgements.**   This project has received funding from the European Research Council (ERC) under the European Union's Horizon 2020 research and innovation programme (grant agreement n[o] 757360). NSD and SH were supported in part by a research grant (15334) from VILLUM FONDEN. We gratefully acknowledge the support of NVIDIA Corporation with the donation of GPU hardware used for this research.

## Footnotes

* Section for Cognitive Systems, Technical University of Denmark

[2]Follows from $\mathcal{T}_{\boldsymbol{\gamma}}$ and $\mathcal{T}_{-\boldsymbol{\gamma}}$ being commuting matrices.

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
