[Supplementary Material]

# Explicit Disentanglement of Appearance and Perspective in Generative Models — Supplementary Material —

**Nicki Skafte Detlefsen  Søren Hauberg**

## Abstract

*This document contains supplementary material for the paper "Explicit Disentanglement of Appearance and Perspective in Generative Models". The current document contains the following sections: 1. Derivation of the ELBO for C-VITAE and U-VITAE, 2. Implementation details for the experiments, 3. Experiments on the robustness of different affine parameterizations, 4. Additional results*

## 1. Derivation of the ELBO for C-VITAE and U-VITAE

We will here focus on deriving the ELBO for the C-VITAE, because as we will see the ELBO for the U-VITAE can easily be identified from this. For both models it hold that the generative model is given by

$$p(\boldsymbol{x}) = \iint p(\boldsymbol{x}|\boldsymbol{z}_A, \boldsymbol{z}_P)p(\boldsymbol{z}_A)p(\boldsymbol{z}_P)\mathrm{d}\boldsymbol{z}_A\mathrm{d}\boldsymbol{z}_P.$$

We know assume that the inference of appearance now becomes dependent on the perspective factors $\boldsymbol{z}_P$ *i.e.*

$$p(\boldsymbol{z}_P|\boldsymbol{x}) \approx q_P(\boldsymbol{z}_P|\boldsymbol{x}) \text{ and } p(\boldsymbol{z}_A|\boldsymbol{x}) \approx q_A(\boldsymbol{z}_A|\boldsymbol{x}, \boldsymbol{z}_P).$$

as in the C-VITAE model. The log-posterior is then given by:

$$
\begin{aligned}
\log p(\boldsymbol{x}) &= \log \left( \iint p(\boldsymbol{x}|\boldsymbol{z}_A, \boldsymbol{z}_P)p(\boldsymbol{z}_A)p(\boldsymbol{z}_P)\mathrm{d}\boldsymbol{z}_A\mathrm{d}\boldsymbol{z}_P \right) \\
&= \log \left( \iint p(\boldsymbol{x}|\boldsymbol{z}_A, \boldsymbol{z}_P)p(\boldsymbol{z}_A)p(\boldsymbol{z}_P) \frac{q_A(\boldsymbol{z}_A|\boldsymbol{z}_P, \boldsymbol{x})}{q_A(\boldsymbol{z}_A|\boldsymbol{z}_P, \boldsymbol{x})} \frac{q_P(\boldsymbol{z}_P|\boldsymbol{x})}{q_P(\boldsymbol{z}_P|\boldsymbol{x})}\mathrm{d}\boldsymbol{z}_A\mathrm{d}\boldsymbol{z}_P \right) \\
&= \log \left( \int \mathbb{E}_{q_A(\boldsymbol{z}_A|\boldsymbol{z}_P, \boldsymbol{x})} \left[ \frac{p(\boldsymbol{x}|\boldsymbol{z}_P, \boldsymbol{z}_A)p(\boldsymbol{z}_A)}{q_A(\boldsymbol{z}_A|\boldsymbol{z}_P, \boldsymbol{x})} \right] p(\boldsymbol{z}_P) \frac{q_P(\boldsymbol{z}_P|\boldsymbol{x})}{q_P(\boldsymbol{z}_P|\boldsymbol{x})}\mathrm{d}\boldsymbol{z}_P \right) \\
&= \log \mathbb{E}_{q_P(\boldsymbol{z}_P|\boldsymbol{x})} \left[ \mathbb{E}_{q_A(\boldsymbol{z}_A|\boldsymbol{z}_P, \boldsymbol{x})} \left[ \frac{p(\boldsymbol{x}|\boldsymbol{z}_P, \boldsymbol{z}_A)p(\boldsymbol{z}_A)}{q_A(\boldsymbol{z}_A|\boldsymbol{z}_P, \boldsymbol{x})} \right] \frac{p(\boldsymbol{z}_P)}{q_P(\boldsymbol{z}_P|\boldsymbol{x})} \right]
\end{aligned}
$$

By using Jensen's inequality once to exchange the outer expectation with the $\log$ gives us

$$
\begin{aligned}
\log p(\boldsymbol{x}) &\geq \mathbb{E}_{q_P(\boldsymbol{z}_P|\boldsymbol{x})} \left[ \log \left( \mathbb{E}_{q_A(\boldsymbol{z}_A|\boldsymbol{z}_P, \boldsymbol{x})} \left[ \frac{p(\boldsymbol{x}|\boldsymbol{z}_P, \boldsymbol{z}_A)p(\boldsymbol{z}_A)}{q_A(\boldsymbol{z}_A|\boldsymbol{z}_P, \boldsymbol{x})} \right] \right) + \log \left( \frac{p(\boldsymbol{z}_P)}{q_P(\boldsymbol{z}_P|\boldsymbol{x})} \right) \right] \\
&= \mathbb{E}_{q_P(\boldsymbol{z}_P|\boldsymbol{x})} \left[ \log \left( \mathbb{E}_{q_A(\boldsymbol{z}_A|\boldsymbol{z}_P, \boldsymbol{x})} \left[ \frac{p(\boldsymbol{x}|\boldsymbol{z}_P, \boldsymbol{z}_A)p(\boldsymbol{z}_A)}{q_A(\boldsymbol{z}_A|\boldsymbol{z}_P, \boldsymbol{x})} \right] \right) \right] - D_{KL}(q_P(\boldsymbol{z}_P|\boldsymbol{x})||p(\boldsymbol{z}_P))
\end{aligned}
$$

Then, by using Jensen's inequality once more to exchange the $\log$ and inner expectation we get

$$\log p(\boldsymbol{x}) \geq \mathbb{E}_{q_P(\boldsymbol{z}_P|\boldsymbol{x})}\left[\mathbb{E}_{q_A(\boldsymbol{z}_A|\boldsymbol{z}_P,\boldsymbol{x})}\left[\log p(\boldsymbol{x}|\boldsymbol{z}_P,\boldsymbol{z}_A) + \log\left(\frac{p(\boldsymbol{z}_A)}{q_A(\boldsymbol{z}_A|\boldsymbol{z}_P,\boldsymbol{x})}\right)\right]\right] - D_{KL}(q_P(\boldsymbol{z}_P|\boldsymbol{x})||p(\boldsymbol{z}_P))$$

$$= \underbrace{\mathbb{E}_{q_P(\boldsymbol{z}_P|\boldsymbol{x})}\left[\mathbb{E}_{q_A(\boldsymbol{z}_A|\boldsymbol{z}_P,\boldsymbol{x})}\left[\log p(\boldsymbol{x}|\boldsymbol{z}_P,\boldsymbol{z}_A)\right]\right]}_{\text{term 1}} - \underbrace{\mathbb{E}_{q_P(\boldsymbol{z}_P|\boldsymbol{x})}\left[D_{KL}(q_A(\boldsymbol{z}_A|\boldsymbol{z}_P,\boldsymbol{x})||p(\boldsymbol{z}_A))\right]}_{\text{term 2}} - \underbrace{D_{KL}(q_P(\boldsymbol{z}_P|\boldsymbol{x})||p(\boldsymbol{z}_P))}_{\text{term 3}}$$

Here term 1 is reconstruction term between $\boldsymbol{x}$ and $p(\boldsymbol{x}|\boldsymbol{z}_A,\boldsymbol{z}_P)$, is the term 2 is the KL divergence for the appearance space $q_A(\boldsymbol{z}_A|\boldsymbol{z}_P,\boldsymbol{x})$ and its prior $p(\boldsymbol{z}_A)$ and term 3 is the KL divergence for the perspective space $q_P(\boldsymbol{z}_P|x)$ and its prior $p(\boldsymbol{z}_P)$. Similar to how gradients are calculate in VAE's, the outer expectation in term 2 is calculated with respect to a single sample, but can also be computed with respect to multiple samples similar to the work of Burda et al. (2015).

To get the ELBO of the U-VITAE model, we make the the inference of the latent spaces independent of each other *i.e.* $q_A(\boldsymbol{z}_A|z_P,x) = q_A(\boldsymbol{z}_A|x)$. This get rid of the expectation in term 2 and we are left with

$$\log p(\boldsymbol{x}) \geq \mathbb{E}_{q_P(\boldsymbol{z}_P|\boldsymbol{x})}\left[\mathbb{E}_{q_A(\boldsymbol{z}_A|\boldsymbol{z}_P,\boldsymbol{x})}\left[\log p(\boldsymbol{x}|\boldsymbol{z}_P,\boldsymbol{z}_A)\right]\right] - D_{KL}(q_A(\boldsymbol{z}_A|\boldsymbol{x})||p(\boldsymbol{z}_A)) - D_{KL}(q_P(\boldsymbol{z}_P|\boldsymbol{x})||p(\boldsymbol{z}_P)),$$

which is the ELBO for the U-VITAE model. The intuition behind this equation is that the U-VITAE model is just a standard VAE, where the latent space $\boldsymbol{z}$ has been split into two smaller latent spaces $\boldsymbol{z}_P, \boldsymbol{z}_A$, thus this is reflected in ELBO where the KL-term is similar split into two terms.

## 2. Implementation details for the experiments

Below we describe the network architectures in details. All models were trained using the Adam optimizer (Kingma & Ba, 2014) with fixed learning rate of $10^{-4}$. For the MNIST experiments we used a batch size of 512 and trained for a 2000 epochs and for SMLP and CelebA experiments we used a batch size of 256 and trained for 5000 epochs. No early stopping was used. Similar to Kaae Sønderby et al. (2016), we use annealing/warmup for the KL-divergence by scaling the term(s) by $w = \min\left(\frac{\text{epoch}}{\text{warmup}}, 1\right)$, where the warmup parameter was set to half the number of epochs.

**Details for MNIST experiments**. Pixel values of the images are scaled to the interval [0,1]. Each pixel is assumed to be Bernoulli distributed. For the encoders and decoders we use multilayer perceptron networks. For the VAE, $\beta$-VAE (Higgins et al., 2017), $\beta$-TCVAE (Chen et al., 2018) and DIP-VAE (Kumar et al., 2017), we use the settings listed below. For both VITAE models, we model both encoders and both decoders with approximately half the neurons, for a fair comparison. In practice we found that the encoders/decoders of the appearance factors benefits from having a bit higher capacity than the encoders/decoders of the perspective factors.

| | Layer 1 | Layer 2 | Layer 3 |
|---|---|---|---|
| $\boldsymbol{\mu}_{encoder}$ | 128, (LeakyReLU) | 64, (LeakyReLU) | d, (Linear) |
| $\boldsymbol{\sigma}^2_{encoder}$ | 128, (LeakyReLU) | 64, (LeakyReLU) | d, (softplus) |
| $\boldsymbol{\mu}_{decoder}$ | 64, (LeakyReLU) | 64, (LeakyReLU) | D, (Sigmoid) |

*Table 1.* Model architecture for the MNIST experiments.

Here $D = 784$ and $d = 4$ for VAE based models and $d = 2$ for VITAE based models. The numbers corresponds to the size of the layer and the parenthesis is the used activation function. For the LeakyRelu activation function we use hyper parameter $\alpha = 0.1$. We only parametrize a mean function in the decoder because we assume the output pixels are Bernoulli distributed.

**Details for SMPL experiments**. Images was generated using the SMPL library[1]. The parameters for generating the body shape was drawn from a $\mathcal{N}(0, 1.25^2)$ distribution. The parameters that controls the body pose was uniformly sampled from one out of 4 pre-specified pose configurations, see Table 2.

The resolution of each image was scaled down to $(400, 200)$. Each pixel is assumed to be Normal distributed. For the VAE

|                | Pose 1     | Pose 2      | Pose 3     | Pose 4     |
| -------------- | ---------- | ----------- | ---------- | ---------- |
| Left shoulder  | $-\pi/8$   | $-\pi/16$   | $\pi/16$   | $\pi/8$    |
| Right shoulder | $\pi/8$    | $\pi/16$    | $-\pi/16$  | $-\pi/8$   |
| Left arm       | $-\pi/3.5$ | $-\pi/3.5$  | $\pi/3.5$  | $\pi/3.5$  |
| Right arm      | $\pi/3.5$  | $\pi/3.5$   | $-\pi/3.5$ | $-\pi/3.5$ |

*Table 2.* When generating synthetic bodies, we uniformly sample one of the above settings for the pose.

based models, we use the settings listed below. For the VITAE models we used approximately half the neurons for the encoders/decoders.

|                          | Layer 1             | Layer 2             | Layer 3       |
| ------------------------ | ------------------- | ------------------- | ------------- |
| $\boldsymbol{\mu}_{encoder}$    | 256, (LeakyReLU)    | 128, (LeakyReLU)    | d, (Linear)   |
| $\boldsymbol{\sigma}^2_{encoder}$ | 256, (LeakyReLU)    | 128, (LeakyReLU)    | d, (softplus) |
| $\boldsymbol{\mu}_{decoder}$    | 128, (LeakyReLU)    | 256, (LeakyReLU)    | D, (Linear)   |

*Table 3.* Model architecture for the SMPL experiments.

Here $D = 80.000$ and $d = 4$ for VAE based models and $d = 2$ for VITAE based models. The numbers corresponds to the size of the layer and the parenthesis is the used activation function. For the LeakyRelu activation function we use hyper parameter $\alpha = 0.1$. We only parametrize a mean in the decoder because the variance function is in general very hard to train and completely arbitrarily outside the latent manifold (Arvanitidis et al., 2017). It was therefore fixed for all pixels in all images to $\sigma^2_{decoder} = 0.1$. For the CPAB transformations (Freifeld et al., 2015) we ran the experiments with tessellation parameters [2, 4] with zero boundary constrains and no volume preservation constrains. With these settings, we are generating perspective transformations of size 30 *i.e.* $\dim(\boldsymbol{\theta}) = 30$.

**Details for CelebA experiments**. We use the align and cropped version of the dataset, downloaded from the homepage[2]. Each image was then down sampled to size $128 \times 128$, to decrease computational time. Each pixel is assumed to be Normal distributed. For this task we use a convolutional-VAE. Below is listed the configuration of the network:

|                          | Layer 1                         | Layer 2                         | Layer 3                         | Layer 4                  |
| ------------------------ | ------------------------------- | ------------------------------- | ------------------------------- | ------------------------ |
| $\boldsymbol{\mu}_{encoder}$    | Conv(10, 5, 2, LeakyReLU)       | Conv(20, 5, 2, LeakyReLU)       | Conv(40, 3, 2, LeakyReLU)       | Dense(2, Linear)         |
| $\boldsymbol{\sigma}^2_{encoder}$ | Conv(10, 5, 2, LeakyReLU)       | Conv(20, 5, 2, LeakyReLU)       | Conv(40, 3, 2, LeakyReLU)       | Dense(2, Softplus)       |
| $\boldsymbol{\mu}_{decoder}$    | DeConv(40, 3, 2, LeakyReLU)     | DeConv(20, 3, 2, LeakyReLU)     | DeConv(10, 5, 2, LeakyReLU)     | DeConv(3, 5, 2, Sigmoid) |

*Table 4.* Model architecture for the CelebA experiments. Conv denotes a convolutional layer and DeConv denotes de-/transposed convolutional layers. The parameters are respective number of filters, filter size, stride and activation function.

For the CPAB transformation (Freifeld et al., 2015) we ran the experiments with tessellation parameters [4, 4] with zero boundary constrains and no volume preservation constrains. With these settings, we are generating perspective transformations of size 62.

**Computational requirements.** Even though VITAE has a more complicated architecture than VAE (comparing Fig. (3a) vs. (3c) in main paper) both forward and backward passes in the models have roughly the same complexity when we use affine transformations (see Table 5). Using the more complex CPAB transformations adds some penalty to the computational time.

## 3. Stability results

In the main paper we discuss multiple ways to parameterize an affine transformation. If we choose $\mathcal{T}_\gamma$ with a diffiomorphic parameterization, we have found that this also has positive positive optimization properties. Fig. 1 shows the ELBO as a function of the learning rate $\lambda$ for the three different choices of affine parametrization discussed in the main paper, using our C-VITAE architecture. We clearly see that the diffeomorphic affine parametrization archives a tighter bound, and can run for much higher learning rates (faster convergence) before the network begins to diverge. These findings are similar to those of Detlefsen et al. (2018) in the supervised context.

|  |  | Forward | Backward |
|---|---|---|---|
| VAE & $\beta$-VAE | | 0.0016s | 0.014s |
| $\beta$-TCVAE | | 0.0020s | 0.016s |
| DIP-VAE-II | | 0.0025s | 0.018s |
| C-VITAE | Affine | 0.0092s | 0.037s |
| | CPAB | 0.1s | 0.86s |

*Table 5.* Forward and backwards timings for the different architectures. The experiments was conducted with an Intel Xeon E5-2620v4 CPU and Nvidia GTX TITAN X GPU.

*Figure 1.* Top: Stability towards choice of learning rate for three different parametrizations of affine transformations. Missing values indicates that the network diverged. Bottom: Learning curves also show that the diffiomorphic affine parametrization converges faster and is more stable in its training.

These experiments was conducted on the MNIST dataset. For all three experiments we use the C-VITAE architecture with a neural network structure as Table 1. A batch size of 512 was used. The results where generated by changing the parametrization of the affine spatial transformer between

Affine
$$\mathcal{T}_\gamma(\boldsymbol{x}) = \begin{bmatrix} \gamma_{11} & \gamma_{12} & \gamma_{13} \\ \gamma_{21} & \gamma_{22} & \gamma_{14} \end{bmatrix} \begin{bmatrix} x \\ y \\ 1 \end{bmatrix} \tag{1}$$

AffineDecomp
$$\mathcal{T}_\gamma(\boldsymbol{x}) = \begin{bmatrix} \cos(\alpha) & -\sin(\alpha) \\ \sin(\alpha) & \cos(\alpha) \end{bmatrix} \begin{bmatrix} 1 & m \\ 0 & 1 \end{bmatrix} \begin{bmatrix} s_x & 0 \\ 0 & s_y \end{bmatrix} + \begin{bmatrix} t_x \\ t_y \end{bmatrix} \tag{2}$$

AffineDiffio
$$\mathcal{T}_\gamma(\boldsymbol{x}) = \mathbf{expm}\left( \begin{bmatrix} \gamma_{11} & \gamma_{12} & \gamma_{13} \\ \gamma_{21} & \gamma_{22} & \gamma_{14} \\ 0 & 0 & 0 \end{bmatrix} \right) \begin{bmatrix} x \\ y \\ 1 \end{bmatrix} \tag{3}$$

and by varying the learning rate $\lambda = \{10^{-4}, 10^{-3}, 10^{-2}, 10^{-1}\}$. The lower subplot of Figure 4, was generated using a learning rate of $\lambda = 10^{-4}$ to make sure that all transformer types would converge.

## 4. Additional results

### 4.1. MNIST experiments

In Fig. 2 reconstructions from the different models can be seen. In Fig. 3 generated sampler from the different models can be seen. In Fig. 4 latent manipulations can be seen.

(a) VAE

(b) $\beta$-VAE

(c) $\beta$-TCVAE

(d) DIP-VAE-II

(e) C-VITAE

*Figure 2.* Samples from the test set (top rows) and the corresponding reconstructions (bottom rows) for all models. We clearly observe that the additional weight on the KL term in $\beta$-VAE, $\beta$-TCVAE and DIP-VAE-II makes the reconstructions worse.

(a) VAE

(b) $\beta$-VAE

(c) $\beta$-TCVAE

(d) DIP-VAE-II

(e) C-VITAE

*Figure 3.* Samples from the prior distribution.

(a) VAE

(b) $\beta$-VAE

(c) $\beta$-TCVAE

(d) DIP-VAE-II

(e) C-VITAE

*Figure 4.* Latent manipulation. The images were generated by varying one latent dimension, while keeping the rest fixed. We choose the latent variable that qualitatively gave the best results.

## 5. SMPL experiment

In Fig. 5 reconstructions from the different models can be seen. In Fig. 6 generated sampler from the different models can be seen. In Fig. 4 latent manipulations can be seen.

(a) VAE

(b) $\beta$-VAE

(c) $\beta$-TCVAE

(d) DIP-VAE-II

(e) C-VITAE

Figure 5. Test set reconstructions on SMPL dataset.

(a) VAE

(b) $\beta$-VAE

(c) $\beta$-TCVAE

(d) DIP-VAE-II

(e) C-VITAE

*Figure 6.* Samples from the prior distribution.

*Figure 7.* Disentanglement of body shape and body pose on SMPL-generated bodies for all models. The images are generated by varying one latent dimension, while keeping the rest fixed. For the C-VITAE model we have shown this for both the appearance and perspective spaces, since this is the only model where we quantitatively observe disentanglement.

## Footnotes

[1] http://smpl.is.tue.mpg.de/

[2]http://mmlab.ie.cuhk.edu.hk/projects/CelebA.html