[Reviews · NeurIPS 2019]

Reviewer 1



* Quality Overall the paper does execute the combination of the spatial transformer layer with the VAE well, but this seems relatively straightforward. My main concern is with the quantitative evaluation. The paper follows Eastwood and Williams 2019 which proposes a framework to evaluate disentanglement. However: 1. Why this approach? What are the other methods? How to measure “disentanglement” seems non-trivial (and "unsolved") and as such, for a paper in this domain, the evaluation method will be critical. 2. The framework consists of 3 scores, of which the used “disentanglement score” is only one of them. Why are the others not used? If there is a good reason, it would be important to mention this. 3. The framework relies on synthetic data, data where the ground-truth latent structure is known and available. But for MNIST these are not known, the paper assumes the generative factors are the class labels, which is quite different from the type of generative factors generally considered: size, position, color, etc. The paper shows similar style variations controlled with the latents qualitatively, but these aren’t “known” and thus its disentanglement of them can’t be measured by this metric. This makes the comparison of this metric for MNIST less relevant. 4. All generative factors considered in Eastwood and Williams are continuous, whereas in this paper they are all discrete; for SMPL they are intentionally discretized whereas in Eastwood and Williams they even argue in the related work that such discretization is unnecessary. 5. The likelihood differences between the different models look very small (Table 3). It would be good to show the variance across different runs and have some measure of significance for these differences. * Clarity The paper was quite easy to follow, and seems well-structured. * Originality The paper mostly combines two well-known methods: VAEs and spatial transformers. There is extensive related work discussed and the approach seems properly positioned with respect to it. However, as noted in the Quality remarks, how to evaluate disentanglement seems non-trivial. Some clearer argument of why the current metric (from Eastwood and Williams) is being used, and what other approaches have used, is important. * Significance I would consider the significance relatively minor, the scope seems too limited to be of wider significance. ---------------------------------- I thank the authors for their response. I do think, as in my original review, that more justification / background for the choice of metrics is important. Currently, there is no discussion about the evaluation itself, only a statement about the chosen metric L195-197, which is insufficient, especially when the technical novelty is relatively minor. If "Locatello et al. (2019, ICML best paper) find that most proposed disentanglement metrics, including the disentanglement score from Eastwood et al. (2018) that we are using, are highly correlated" -- also noting the "most" -- why not report more metrics and provide more evidence for that (at the very least in the appendix)? In Locatello et al. they conclude that the level of correlation across metrics depends on the dataset, such that I don't think we can conclude that there will be high correlation for _all metrics_ and _all_ datasets (e.g. CelebA which isn't investigated in Locatello et al.). Regardless, it seems premature to pick and fix a single metric for such a notoriously hard to measure task (with many different interpretations), even if there is very recent evidence that _most of_ the scores are correlated _for the current (class of) models_ on some of the current datasets. This seems similar to the difficulty of measuring sample quality and/or diversity in generative models (e.g. [1]). For generative models Inception Score was, and still is, used to provide some measure of sample quality / diversity of GANs, and it qualitatively seemed correlated with improved results. However, it clearly has multiple flaws, see [2], and simply seemed useful at the time as some proxy of progress within GANs but broke down in certain regimes / model classes. Similarly, IS and Fréchet Inception Distance (FID) seem quite correlated with progress in visual sample quality, but do behave quite differently in some settings (see e.g. [3]). Anecdotally, although somewhat discussed in [1], improvements in likelihood of generative models is correlated with improvements in sample quality, when measured within the same model class. Once you try to compare different models you can get nonsense results. Regarding "We focus on the disentanglement score as it is the only one that seems to be of general interest in the field", this seems unconvincing given how only very recently (Locatello et al.) there has been some evidence that "most of the metrics" are correlated (with different levels), as discussed above. I feel we should be much more careful just immediately dismissing any other metrics on such a hard-to-measure task. However, I appreciate the authors have done more experiments on more datasets, and addressed some of my comments. I am still unsatisfied with the evaluation and the somewhat dismissive rebuttal for that, such that I'm only changing my score to 5. Commitment to more discussion about the evaluation (with reference to e.g., Locatello et al., which is currently only referenced in different contexts) and more evaluation results with more metrics, feels critical to me and would have made me satisfied enough to give a higher score. [1]: Theis et al., A note on the evaluation of generative models, https://arxiv.org/abs/1511.01844 [2]: Barrat & Sharma, A Note on the Inception Score, https://arxiv.org/abs/1801.01973 [3]: Heusel et al., GANs Trained by a Two Time-Scale Update Rule Converge to a Local Nash Equilibrium, https://arxiv.org/abs/1706.08500

Reviewer 2



The need for the transformer network in the encoder is not clearly explained. Can you achieve better disentanglement by discouraging perspective code from predicting appearance code? Would Wasserstein auto encoder be a more powerful framework than VAE? Below are two related papers: 1. Unsupervised Part-Based Disentangling of Object Shape and Appearance Dominik Lorenz, Leonard Bereska, Timo Milbich, Björn Ommer 2. Unsupervised Disentangling of Appearance and Geometry by Deformable Generator Network Xianglei Xing, Tian Han, Ruiqi Gao, Song-Chun Zhu, Ying Nian Wu; The IEEE Conference on Computer Vision and Pattern Recognition (CVPR), 2019, pp. 10354-10363 ----after author feedbacks--- The authors provided results on more datasets, dSprites and CelebA. The authors also clarified their contributions with respect to related papers (transformations in related papers are not invertible and hence do not correspond to proper group actions). The authors also commented on WAE vs. VAE. My main concerns are addressed to my satisfaction. Therefore, I am happy to increase my rating to 6. The reason that my rating is not higher is that, the framework is rather straightforward technically. For more technical depth, the authors should consider analyzing the use of inverse transform in encoder from a mutual information perspective. This key contribution needs to be backed up by rigorous theoretical analysis.

Reviewer 3



Originality: This paper proposes to combine VAEs and spatial transformer networks (STNs) to learn a disentangled representation of appearance and perspective. While STNs have been combined with generative models before (e.g. AIR by Eslami et al, 2016), to my knowledge this is the first time this approach has been used for learning disentangled representations. Quality: The paper appears technically sound and the results are supported by both quantitative and qualitative measures. Clarity: This is one of the most well written papers I have read recently. Significance: This paper makes an important step towards a new direction towards unsupervised disentangled representation learning, where transformations take the centre stage. This is in line with the recent theoretical work that suggests defining disentangled representations in terms of symmetry transformations (Higgins et al, 2018). While the current implementation with STNs is limited to affine transforms, the setup is general enough to replace STNs with more powerful transformation learning models. I believe that others are likely to build on top of this work in the future. ----- After author feedback ------ The additional experiments provided by the authors in the rebuttal have made me even more happy to support the acceptance of this paper. I keep my original high score unchanged but argue for the acceptance.

Reviewer 4



Strength: Architecture seems to be novel. Introduction of spatial transformer to represent perspective with a VAE as well as the use of perspective inference to better obtain disentangled appearance. Weakness: -Experiments are performed on two datasets: MNIST and SMPL. Both datasets are toy-like and quite different from typical images. The MNIST task is too simple. -Experimental results are somewhat promising but not convincing. More complex / diverse datasets would provide evidence that this is good method to disentangle appearance and perspective. -Fails to achieve high quality disentanglement on one baseline. Comments: -In C-VITAE the encoder for z_A explicitly depends on z_P, so it is a little confusing how it can be that conditioning can help with disentangling z_A from z_P. Since the prior for z_A is not conditioned on z_P (for C-VITAE), it will have the effect of minimizing mutual information between z_A and z_P. Is this important to achieving disentanglement? -Typo: DLR -> DRL twice on page 2. Originality: Architecture is original. Clarity: Very clear. Significance: Unclear as the experimental results are not 100% convincing.

[Author Response · NeurIPS 2019]

## NeurIPS Rebuttal for "Explicit Disentanglement of Appearance and Perspective in Generative Models"

We thank the reviewers for their constructive and fair reviews. We here address their key concerns. Our main contribution to NeurIPS is the first practical realization of the theory from Higgins et al. (2017), which argue that disentanglement need to appear from *group actions*. Our work is, thus, an important first-step towards bridging theory and practice. All comments regarding text+figure updates are appreciated and have been added to the paper.

**Experiments(R1+R2+R5):** As requested by multiple reviewers, we empirically evaluated our proposed VITAE model on more datasets. Specifically, the paper now include performance on the dSprites (Higgins et al., 2017) dataset (quantitative evaluation, see Table 1) and qualitative disentanglement of facial attributes on CelebA dataset. Figure 1 illustrates how our model is able to capture geometrical face information such as the shape of the face. We found similarly that we could also capture head pose and viewing angle.

|        | $D_{score}$ |
|--------|-------------|
| VAE    | 0.05        |
| $\beta$-VAE | 0.18   |
| $\beta$-TCVAE | 0.30 |
| DIP-VAE-II | 0.12    |
| C-VITAE | **0.38**   |

Table 1: dSprites

Figure 1: CelebA

**Encoder network (R2+R5):** There is concern about the importance of making the appearance encoder dependent on the perspective through the inverse transformation. We stress that this choice is crucial, and constitutes a notable part of our contribution. Empirically we find that C-VITAE (with encoder-transformer) outperforms U-VITAE (without encoder-transformer) in all tests, but did only include the results in the case of MNIST. As R5 notes, this conditioning minimizes the mutual information between $z_A$ and $z_P$, leading to better disentanglement of the two latent spaces. Thus, there is no need to directly discourage the perspective factors from predicting the appearance factors by for example including an auxiliary loss term. This is rather elegant. Lastly, by forcing the encoder-transformer to be the inverse of the decoder-transformer we mimic the behaviour of normalizing flow models, which have better inference properties.

**R1:** We agree that measuring "disentanglement" is an unsolved problem, but Locatello et al. (2019, ICML best paper) find that most proposed disentanglement metrics, including the disentanglement score from Eastwood et al. (2018) that we are using, are highly correlated. Hence, our results should still hold under other established metrics. We focus on the disentanglement score as it is the only one that seems to be of general interest in the field.

While the underlying generative factors are not known for MNIST in the sense as for synthetic data, we do not agree that the label of an image cannot be interpreted as a generative factor. If we ever are to move away from only quantitatively evaluating disentanglement on synthetic data, we need to consider which generative factors real life data could have been generated from. We agree that the generative factors for SMPL where made discrete on purpose, and we have adjusted this in the revised paper. This discretization does not invalidate our results, since the general framework from Eastwood et al. (2018) still holds. Secondly, when Eastwood et al. (2018) state that "discretization is unnecessary" they comment on the discretization of latent factors and not on discretization of generative factors, as in our case. Regarding significance, we re-ran all algorithms three times, and can conclude the results indeed are significant.

**R2:** Indeed, it should be possible to exchange the VAE framework with a WAE, by essentially changing the loss function. We do not expect better disentanglement, but rather improved reconstructions as in the original WAE article.

We agree that the work of Lorenz et al. (2019) and Xing et al. (2019) are related to ours in that they rely on spatial transformations to disentangle data (note: both articles were published at CVPR 2019 after the NeurIPS deadline). That said, the theory of Higgins et al., strongly relies on disentanglement through *group actions*. This places hard constraints on which spatial transformations are allowed: *they have to form a smooth group*. Both TPS-transformation (Lorenz et al.) and displacement fields (Xing et al.) are not invertible and hence do not correspond to proper group actions. Our work, thus, remains unique as the first realization of the theory from Higgins et al. (2017). As a practical benefit, we have also shown that the invertibility of the group action lead to efficient inference through the C-VITAE architecture.

**R4:** We thank you for the positive review. We agree that even if our approach is novel, it can always be improved upon. We want to stress that the results on the SMPL dataset actually take advantage of the CPAB transformations. We made this more clear in the final version.

**R5:** We do not agree that our SMPL dataset is toy-like, which should be evident from the fact that even state-of-the-art models seem to struggle with disentanglement of the generative factors. Additionally, they are in the same level of complexity as the dSprites, Cars3D, SmallNORB and Shapes3D datasets that are currently used for disentanglement evaluation, see Locatello et al. (2019). We choose explicitly to work on datasets where we have access to generative factors, such that our results could be quantitatively evaluated. This however heavily limits the datasets to be either synthetic or very simple in complexity.

[Meta-Review · NeurIPS 2019]

The paper that tries to disentangle appearance from perspective based on a variational autoencoder with a spatial transformer layer. The paper proposes a novel architecture and, as one of the first empirical works, builds on recent work of linking disentangled representations with symmetry transformations. Several reviewers appreciated the author response that also contained new experiments. The original experiments were based on mostly simplistic datasets, but the authors convincingly provided more experimental results during the rebuttal process. The evaluation could further be improved by adopting other metrics as pointed out by R1.